

# Molecular epidemiology of vancomycin-resistant *Enterococcus faecium* clinical isolates in a tertiary care hospital in southern Thailand: a retrospective study

Phanvasri Saengsuwan[1], Kamonnut Singkhamanan[1], Siribhorn Madla[2], Natnicha Ingviya[3] and Chonticha Romyasamit[4]

[1] Department of Biomedical Sciences and Biomedical Engineering, Faculty of Medicine, Prince of Songkla University, Hatyai, Songkhla, Thailand
[2] School of Pharmacy, Walailak University, Thasala, Nakhon Si Thammarat, Thailand
[3] Department of Pathology, Faculty of Medicine, Prince of Songkla University, Hatyai, Songkhla, Thailand
[4] School of Allied Health Sciences, Walailak University, Thasala, Nakhon Si Thammarat, Thailand

Corresponding author
Phanvasri Saengsuwan,
sphanvas@medicine.psu.ac.th

## ABSTRACT

**Objective**. Vancomycin-resistant enterococci are nosocomial pathogens that are responsible for commonly causing healthcare-associated infections, and they exhibit increased resistance to many antimicrobials, particularly to vancomycin. The epidemiological data available on vancomycin-resistant enterococci (VRE) in Thailand are inadequate.

**Methods**. Using enterobacterial repetitive intergenic consensus-polymerase chain reaction (ERIC-PCR), this study investigated genes that encode antimicrobial resistance and genetic relatedness to further understand VRE prevalence. Ninety VRE isolates were collected between 2011 and 2019 from a tertiary care hospital in southern Thailand. Antimicrobial susceptibility was determined using the disk diffusion method and E-test methods. Multiplex PCR was performed to detect the *van* gene and virulence genes.

**Results**. The study showed a high prevalence of diverse multidrug-resistant VRE strains. The prevalence of VRE infection was the highest in 2014 (28 isolates, 39.4%). VRE were mostly found in the urogenital tract (26 isolates, 28.9%), followed by the digestive tract (20%), body fluid, i.e., pancreatic cyst fluid, peritoneal dialysis fluid, Jackson–Pratt (JP) drain (20%), and blood specimens (10%). Patients in medical and surgical wards had 71.1% multi-drug-resistant and 28.9% extensively drug-resistant (XDR) VRE strains, respectively. The most prevalent antibiotic resistance was to ampicillin (74.4%). Susceptibility to gentamicin and meropenem were similar (7% and 10%, respectively). Four isolates (4.4%) were resistant to colistin. Only *vanA* was detected among the strains. The virulence gene test showed that the detection rates of enterococcal surface protein (*esp*) and hyaluronidase (*hyl*) genes were 91.1% and 5.6%, respectively. According to ERIC-PCR analysis, 51 of 90 strains had clonality, with a similarity rate of 95%.

**Conclusions**. We conclude that there is a need to implement infection control practices and active surveillance. Molecular techniques can effectively detect antibiotic-resistant genes, which would allow monitoring to control VRE infection in hospitals.

## INTRODUCTION

Vancomycin-resistant enterococci (VRE) are nosocomial pathogens that are responsible for causing a large number of hospital-acquired infections worldwide. These organisms affect the bloodstream, urinary tract, and surgical wounds and cause intra-abdominal and intra-pelvic abscesses (*Reyes, Bardossy & Zervos, 2016*). Hospitalized patients who have been using certain antibiotics or are on dialysis for a long time are at an increased risk for VRE colonization and infections (*Arias, Contreras & Murray, 2010*; *Howden et al., 2013*). Enterococci VRE are resistant to commonly used antibiotics, particularly aminoglycoside, ampicillin, vancomycin, and others that affect cell walls (*Billington et al., 2014*; *Puchter et al., 2018*).

With the increase in VRE infections among hospitalized patients, the control of VRE prevalence in healthcare units has become a significant concern for public health experts and for our hospital.

*Enterococcus faecium* and *E. faecalis* are the most common enterococcal species causing nosocomial infections (*Reyes, Bardossy & Zervos, 2016*). According to Thailand's National Antimicrobial Resistance Surveillance Center, the prevalence of *E. faecium* increased from 0.8% to 9.9% between 2012 and 2019, but the occurrence of *E. faecalis* has been stable for 10 years (0.4% in 2018) (*NARST, 2020*).

Using molecular techniques, genes associated with enterococcal virulence can be used for identifying, controlling, and preventing *E. faecium* and *E. faecalis* transmission, as opposed to using biochemical tests that are expensive and time-consuming (*Bhatt et al., 2015*; *Fang et al., 2012*; *Ulu-Kilic et al., 2016*). These virulence factors include *asa* (aggregation substance), *gel* (gelatinase), *cyt* (cytolysin), *esp* (enterococcal surface protein), *hyl* (hyaluronidase), *cpd* (sex pheromone determinant), and *ebp* (endocarditis and biofilm-associated pilus subunit A).

*Esp* from *E. faecalis* contributes to biofilm formation and colonization (*Pillay, Zishiri & Adeleke, 2018*), and *hyl* influences nasopharynx and lung inflammation (*Jackson, Fedorka-Cray & Barrett, 2004*). The molecular techniques used for the identification of *Enterococcus* species are multilocus sequence typing, pulsed-field gel electrophoresis (PFGE) (*Nasaj et al., 2016*), and enterobacterial repetitive intergenic consensus-polymerase chain reaction (ERIC-PCR), which is a simplified typing method for hospital-based epidemiology (*Shahi et al., 2020*).

No reports are available on the molecular analysis or epidemiology of *E. faecium* and *E. faecalis*, and there are limited data on the possible relationship between the prevalence of virulence. This study investigates the incidence rate and genetic relationship of *E. faecium* and *E. faecalis*. We analyze the prevalence of resistance genes using clinical samples collected from a tertiary care hospital in southern Thailand during an 8-year period.

## MATERIALS & METHODS

### Study design and clinical isolates

This retrospective study was conducted at the Songklanagarind Hospital, a 1,000-bed tertiary care hospital in Songkhla province, southern Thailand. The clinical samples used

**Table 1 Data on nosocomial infections due to Enterococci for 2011 to 2019 per year from a tertiary care, Songklanagarind Hospital.**

| Year | Total number (N) of nosocomial enterococcal infections | Number of VRE infections, n (% (n/N), proportion in%) |
|---|---|---|
| 2011 | 1196 | 5 (6, 0.4) |
| 2012 | 1132 | 16 (18, 1.4) |
| 2013 | 839 | 12 (13, 1.4) |
| 2014 | 645 | 28 (31, 4.3) |
| 2015 | 1252 | 5 (6, 0.4) |
| 2016 | 1195 | 5 (6, 0.4) |
| 2017 | 1325 | 3 (3, 0.2) |
| 2018 | 810 | 8 (9, 1.0) |
| 2019 | 879 | 8 (9, 0.9) |
| **Total** | **9273** | **90 (100, 10.5)** |

in this study were acquired from patients between February 2011 and February 2019. Nosocomial enterococci and VRE were identified using conventional biochemical tests at the microbiology laboratory in the hospital by referring to guidelines from the Clinical and Laboratory Standards Institute (*CLSI, 2019*). Ninety VRE isolates were included in this study: five isolates (6%) were recovered in 2011; 16 (18%) in 2012; 12 (13%) in 2013; 28 (31%) in 2014; five (6%) in 2015; five (6%) in 2016; three (3%) in 2017; eight (9%) in 2018; and eight (9%) in 2019 (Table 1).

## Specimen collection and biochemical identification of enterococci species

Specimen types included blood, body fluid (e.g., pancreatic cyst fluid, peritoneal dialysis fluid, and Jackson–Pratt (JP) drain), urine, midstream urine, pus, and samples from the pelvic region, rectum, and tissue. Before use in experiments, all VRE isolates were preserved in 20% glycerol at −80 °C. Sampling of specimen was granted an exemption from requiring ethics approval by the ethical committee of the Faculty of Medicine, Prince of Songkhla University (REC-60-234-04-7).

## Antimicrobial susceptibility tests

Antimicrobial susceptibility tests were performed using the disk diffusion method according to CLSI guidelines (*CLSI, 2019*). Each disk (Becton Dickinson, Heidelberg, Germany) contained ampicillin (AM, 10 µg), cefoperazone-sulbactam (sulperazone) (SPZ, 75/30 µg), cefotaxime (CTX, 30 µg), ceftazidime (CAZ, 30 µg), ceftriaxone (CRO, 30 µg), ciprofloxacin (CIP, 5 µg), colistin (DA, 10 µg), ertapenem (ETP, 10 µg), gentamicin (GM, 10 µg), imipenem (IMP, 10 µg), meropenem (MEM, 10 µg), norfloxacin (NOR, 10 µg), penicillin (P, 10 µg), tazocin (TZP, 100/10 µg), and vancomycin (VA, 30 µg).

Susceptibility testing results were classified as sensitive, intermediate, and resistant according to CLSI breakpoints. Multidrug-resistant (MDR) strains were defined as strains that were resistant to one or more agents in three or more antimicrobial categories

(*Magiorakos et al., 2012*). The minimum inhibitory concentration (MIC) levels for ampicillin-resistant and vancomycin-resistant strains are presented as MIC $\geq$ 16 $\mu$g/mL and MIC $\geq$ 8 $\mu$g/mL, respectively.

## Identification of *Enterococcus faecium* and *E. faecalis* at the species level by multiplex PCR

The genes encoding D-alanine ligases specific for *E. faecium* (*ddl E. faecium*) and *E. faecalis* (*ddl E. faecalis*) and 16S rRNA genes (*rrs*) were detected by a modified multiplex PCR technique using the primers in Table 2 (*Dutka-Malen, Evers & Courvalin, 1995*). The process used was initial denaturation at 95 °C for 10 min, denaturation at 95 ° C for 45 s, annealing at 54 °C for 1 min, and extension at 72 °C for 1 min (35 cycles) (*Bhatt et al., 2015*). The PCR products were visualized on 1.0% agarose gel.

## DNA extraction

Ninety isolates of VRE originating from specimens in a tertiary care teaching hospital (Songkhla, Thailand) were extracted for chromosomal DNA using a GF-1 bacterial DNA extraction kit (Vivantis Technologies Sdn. Bhd., Malaysia) in accordance with the manufacturer's instructions. DNA concentrations were quantified using a spectrophotometer at an absorbance of 260 nm ($A_{260}$). DNA purity was calculated from the ratio of $A_{260}$ and $A_{280}$, and DNA quality was evaluated using agarose gel electrophoresis.

## Detection of *van* genes and virulence genes

All isolates resistant to vancomycin were subjected to a multiplex PCR analysis to detect vancomycin-resistant and virulence genes using a specific primer as detailed in Table 2 (*Bhatt et al., 2015*; *Bourgogne et al., 2007*; *Elsner et al., 2000*; *Vankerckhoven et al., 2004*). PCR amplification was performed via initial denaturation for 3 min at 94 °C; 35 cycles of amplification consisting of 1 min at 94 °C, 1 min at 54 °C, and 1 min at 72 °C; and a final extension for 5 min at 72 °C. A 100-bp DNA ladder (*GeneDireX*, Germany) was used as a molecular-size marker.

ATCC 700221 (for *vanA*) and *E. faecalis* ATCC 51299 (for *vanB*) were used as positive controls. The PCR product was sent for sequencing (1st BASE DNA Sequencing Services, Selangor, Malaysia), and Basic Local Alignment Search Tool search was performed using the National Center for Biotechnology Information database.

## Genotyping by enterobacterial repetitive intergenic consensus (ERIC)-PCR

An ERIC-PCR typing was performed on enterococci strains using the protocol described by *Versalovic, Koeuth & Lupski (1991)* with some modifications. ERIC-1 (ATGTAAGCTCCTGGGGATTCAC) and ERIC-2 (AAGTAAGTGACTGGGGTGAGCG) were the primers. Each PCR reaction (25 $\mu$L) contained 10$\times$ buffer (2.5 $\mu$L), 0.5 mmol/L dNTPs (1 $\mu$L), 5 U/$\mu$L Taq DNA polymerase (0.2 $\mu$L), 100 mmol/L MgCl2 (0.75 $\mu$L), 100 mol/L primers (each 0.25 $\mu$L), template genomic DNA (2 $\mu$L), and distilled water (add to 25 $\mu$L). Amplifications were performed using a C1000 Touch Thermal Cycler (Bio-Rad Laboratories) under the following temperature profiles: initial denaturation at 95 ° C

**Table 2  List of oligonucleotide primers used in the genetic profiling of resistance genes and virulence genes among the isolates in this study.**

| Gene | Primer name | Oligonucleotide sequence (5′ to 3′) | Product size (bp) | Reference |
|------|-------------|-------------------------------------|-------------------|-----------|
| *E. faecalis* | *ddl_ E. faecalis* | ATCAAGTACAGTTAGTCT ACGATTCAAAGCTAACTG | 941 | *Dutka-Malen, Evers & Courvalin (1995)* |
| *E. faecium* | *ddl_ E. faecium* | TTGAGGCAGACCAGATTGACG TATGACAGCGACTCCGATTCC | 658 | |
| *rrs* | 16S rRNA _F | GGATTAGATACCCTGGTAGTCC | 320 | |
| | 16S rRNA_R | TCGTTGCGGGACTTAACCCAAC | | |
| *vanA* | vanA+ | GGGAAAACGACAATTGC | 732 | *Bhatt et al. (2015)* |
| | vanA- | GTACAATGCGGCCGTTA | | |
| *vanB* | vanB+ | ACGGAATGGGAAGCCGA | 647 | |
| | vanB- | TGCACCCGATTTCGTTC | | |
| *vanC* | vanC+ | ATGGATTGGTAYTKGTAT | 815/827 | |
| | vanC- | TAGCGGGAGTGMCYMGTAA | | |
| *vanD* | vanD+ | TGTGGGATGCGATATTCAA | 500 | |
| | vanD | TGCAGCCAAGTATCCGGTAA | | |
| *vanE* | vanE+ | TGTGGGATCGGAGCTGCAG | 430 | |
| | vanE- | ATAGTTTAGCTGGTAAC | | |
| *vanG* | vanG+ | CGGCATCCGCTGTTTTTGA | 941 | |
| | vanG- | GAACGATAGACCAATGCCTT | | |
| *asa* | ASA 11 | GCACGCTATTACGAACTATGA | 373 | *Vankerckhoven et al. (2004)* |
| | ASA 12 | TAAGAAAGAACATCACCACGA | | |
| *gel* | GEL 11 | TATGACAATGCTTTTTGGGAT | 213 | |
| | GEL 12 | AGATGCACCCGAAATAATATA | | |
| *cyt* | CYT I | ACTCGGGGATTGATAGGC | 688 | |
| | CYT IIb | GCTGCTAAAGCTGCGCTT | | |
| *esp* | ESP 14F | AGATTTCATCTTTGATTCTTGG | 510 | |
| | ESP 12R | AATTGATTCTTTAGCATCTGG | | |
| *hyl* | HYL n1 | ACAGAAGAGCTGCAGGAAATG | 276 | |
| | HYL n2 | GACTGACGTCCAAGTTTCCAA | | |
| *cpd* | cpd-F | TGGTGGGTTATTTTTCAATTC | 782 | *Elsner et al. (2000)* |
| | cpd-R | TACGGCTCTGGCTTACTA | | |
| *ebp* | ebpA-F | AAAAATGATTCGGCTCCAGAA | 101 | *Bourgogne et al. (2007)* |
| | ebpA-R | TGCCAGATTCGCTCTCAAAG | | |

for 5 min; 35 cycles of 1 min at 95 °C, 1 min at 48 °C, and 1 min at 72 °C; and a final extension at 72 °C for 10 min. The ERIC-PCR products were separated by electrophoresis in a 1.0% agarose gel with ViSafe green gel stain (0.001%, v/v; Vivantis Technologies Sdn. Bhd., Malaysia) and visualized using the Gel Doc^TM XR ^+ system (Bio-Rad Laboratories, Hercules, CA, US). The images were captured in a TIFF file for further analysis.

ERIC-PCR patterns were analyzed using BioNumerics 7.0 software (Applied Maths, Sint-Martens-Latem, Belgium). A similarity matrix was estimated using Dice's coefficient, and a dendrogram was created based on the unweighted-pair group method with arithmetic averages. Enterococci isolates with a similarity coefficient $\geq$85% were considered as the same genotype (*Said & Abdelmegeed, 2019*).

## Statistical analysis

Demographic data are presented as percentages (unless otherwise stated) and median values with interquartile ranges (IQR). All statistical analyses were performed using SPSS Statistics version 23 (IBM Corporation, Armonk, New York, US). Qualitative variables were compared using a $t$-test and Pearson's chi-squared test. A $P$-value <0.050 was considered statistically significant.

## Ethical approval

This study protocol was approved by the Ethics Committee of the Faculty of Medicine, Prince of Songkhla University (Date: August 4, 2017; Approval ID: REC-60-234-04-7).

## RESULTS

### Characteristics of the study population and VRE sources

Of the 90 VRE isolates recovered from patients, 53 (58.9%) were from females and 37 (41.1%) were from males, for a female-to-male ratio of 3:2. Fifty percent of VRE positive patients were ≥65 years old, and the mean age was 64 years (IQR, 43.5–73; range, 3 months to 90 years). Among these isolates, 42 (46.7%) were obtained from the medical ward; 18 (20.0%) from the surgical ward; and 10 (11.1%) from the intensive care unit. Urine samples had the highest percentage of VRE of 28.9% (26), followed by those obtained from the rectal area at 20% (18) and body fluid at 20% (18). The period of collective sample and source of samples showed statistically significant difference with an antimicrobial pattern ($P < 0.05$), but no significant difference was observed in gender, age groups, and hospital units. The pooled prevalence of VRE infections was 9.6% (95% CI [3.948–16.052]). More epidemiological data are presented in Table 3.

### Antimicrobial resistance profiles

Each VRE isolate was tested for susceptibility to different classes of antimicrobials (sterile fluid, such as blood, and nonsterile sources, such as pus and surgical sites). Of the total 90 VRE isolates, 67 (74.4%) were tested against ampicillin; 57 (63.3%) against imipenem; 21 (23.3%) against gentamicin; 10–20 (11.1%–12.2%) against tazocin, ceftriaxone, cefotaxime, ceftazidime, cefoperazone-sulbactam, meropenem, penicillin, and vancomycin; and less than 10 (<11.1%), against colistin, norfloxacin, ciprofloxacin, and ertapenem.

On the basis of the diameter of the observed zone of inhibition, the result of antibiotic resistance showed that all the selected VRE isolates were found to be resistant to ampicillin, vancomycin, penicillin, tazocin, norfloxacin, and ciprofloxacin susceptibility. Of the 57 selected VRE isolates, 56 (98.2%) were resistant to imipenem. However, over 50% of selected VRE isolates exhibited resistance when tested for gentamicin, cefoperazone-sulbactam, ceftazidime, cefotaxime, ceftriaxone, colistin, and ertapenem (Table 4).

The determination of MIC levels for vancomycin to VRE isolates showed that all the strains of VRE were resistant to vancomycin with MIC ranging between 0.25 µg/ml and 256 µg/ml. Seventy-three isolates had MIC of ≥ 256 µg/ml, and seven had MIC of 128 µg/ml. One isolate was in the intermediate range with the MIC ranging between 8 µg/ml and 16 µg/ml (Table 5).

**Table 3** Demographic and clinical characteristics of patients with VRE at Songklanagarind Hospital between February 2011 and February 2019.

| Variables | MDR-VRE *n* (%) | XDR-VRE *n* (%) | Total (%) | P value |
|---|---|---|---|---|
| *Gender* | | | | 0.883 |
| Male | 26 (40.6) | 11 (42.3) | 37 (41.1) | |
| Female | 38 (59.4) | 15 (57.7) | 53 (58.9) | |
| *Age in years* | | | | 0.669 |
| ≤12 | 3 (4.7) | 0 | 3 (3.3) | |
| 13–24 | 4 (7.8) | 2 (7.7) | 6 (6.7) | |
| 25–64 | 25 (57.8) | 12 (46.2) | 37 (41.1) | |
| ≥65 | 32 (60.9) | 12 (46.2) | 44 (48.9) | |
| *Source of samples* | | | | 0.007* |
| Blood | 7 (10.7) | 2 (7.7) | 9 (10.0) | |
| Body fluid | 10 (19.0) | 8 (30.8) | 18 (20.0) | |
| Midstream urine | 6 (10.7) | 3 (11.5) | 9 (10.0) | |
| Pelvic | 0 | 1 (3.8) | 1 (1.1) | |
| Pus | 0 | 1 (3.8) | 1 (1.1) | |
| Rectal | 18 (21.4) | 0 | 18 (20.0) | |
| Tissue | 3 (7.1) | 5 (19.2) | 8 (8.9) | |
| Urine | 20 (29.8) | 6 (23.1) | 26 (28.9) | |
| *Hospital unit* | | | | 0.353 |
| Medical ward | 31 (48.4) | 11 (42.3) | 42 (46.7) | |
| Gynecology ward | 6 (9.4) | 2 (7.7) | 8 (8.9) | |
| Intensive care unit | 5 (7.8) | 5 (19.2) | 10 (11.1) | |
| Surgical ward | 15 (23.4) | 3 (11.5) | 18 (20.0) | |
| Operating room | 4 (6.3) | 4 (15.4) | 8 (8.9) | |
| Orthopedic ward | 1 (1.6) | 1 (3.8) | 2 (2.2) | |
| Pediatric ward | 2 (3.1) | 0 | 2 (2.2) | |

Notes.

MDR-VRE: multidrug-resistant vancomycin resistant *enterococci*; XDR-VRE: extensively drug-resistant vancomycin resistant *enterococci*.

*$P < 0.05$ is significant.

## Detection of van determinants and virulence factor genes in VRE

Ninety VRE strains (100%) harbored the *vanA* gene, whereas the *vanB*, *vanC*, *vanD*, *vane,* and *vanG* genes were not found in any of these isolates. The *esp* and *hyl* genes were found in 82 (91.1%) and 5 (5.6%) isolates, respectively. However, *gel*, *cyt*, *cpd*, and *ebp* genes were absent from all isolates. The amplicons of the VRE isolates and resistance-gene distribution are shown in Fig. 1.

## ERIC-PCR analysis

The ERIC-PCR patterns of the 90 VRE isolates are shown in Fig. 2. The number of bands in each varied from one to nine, and the ERIC fragment sizes ranged from 100 bp to 1.5 kb. Genomic typification by ERIC-PCR revealed that 51 VRE isolates (56.7%) were grouped into 14 clusters (A–N) at a similarity level of 95%. The predominant A genotype contained 12 isolates; B genotype, six isolates; C–E genotypes, four isolates; F–H genotypes, three

## Table 4 List of antibiotics used in this study.

| Mode of action | Class | No. of tested isolates, *n* (%) | Susceptibility testing results | | |
|---|---|---|---|---|---|
| | | | Sensitive | Intermediate | Resistant |
| Cell membrane | | | | | |
| Colistin | Lipopeptides | 8 (8.9) | 4 | 0 | 4 |
| Cell wall synthesis | | | | | |
| Ampicillin | $\beta$-lactams (penicillins) | 67 (74.4) | 0 | 0 | 67 |
| Penicillin G | $\beta$-lactams (penicillins) | 17 (18.9) | 0 | 0 | 17 |
| Ertapenem | $\beta$-lactams (carbapenem) | 2 (1.1) | 1 | 0 | 1 |
| Imipenem | $\beta$-lactams (carbapenem) | 57 (63.3) | 1 | 0 | 56 |
| Meropenem | $\beta$-lactams (carbapenem) | 15 (16.7) | 10 | 0 | 5 |
| Cefoperazone | $\beta$-lactams (cephalosporins), 3rd generation | 14 (15.6) | 3 | 0 | 11 |
| Cefotaxime | $\beta$-lactams (cephalosporins), 3rd generation | 13 (14.4) | 2 | 0 | 11 |
| Ceftazidime | $\beta$-lactams (cephalosporins), 3rd generation | 14 (15.6) | 2 | 0 | 12 |
| Ceftriaxone | $\beta$-lactams (cephalosporins), 3rd generation | 12 (13.3) | 2 | 0 | 10 |
| Tazocin | Combinations: piperacillin ($\beta$-lactams) and tazobactam ($\beta$-lactamase inhibitors) | 12 (13.3) | 0 | 0 | 12 |
| Vancomycin | Glycopeptides | 20 (22.2) | 0 | 0 | 20 |
| DNA gyrase | | | | | |
| Ciprofloxacin | Fluoroquinolone | 2 (2.2) | 0 | 0 | 2 |
| Norfloxacin | Fluoroquinolone | 5 (5.6) | 0 | 0 | 5 |
| Protein synthesis (30S ribosomal subunit) | | | | | |
| Gentamicin | Aminoglycosides | 21 (23.3) | 7 | 1 | 13 |

## Table 5 Vancomycin MIC results among 90 VRE isolates.

| vancomycin ($\mu$g/ml) | No. of infection | Percent of infection (%) |
|---|---|---|
| 0.75 | 2 | 2.2 |
| 4 | 2 | 2.2 |
| 16 | 1 | 1.1 |
| 24 | 0 | 0 |
| 32 | 0 | 0 |
| 48 | 1 | 1.1 |
| 64 | 3 | 3.3 |
| 96 | 1 | 1.1 |
| 128 | 7 | 7.8 |
| 256 | 53 | 58.9 |
| >256 | 20 | 22.2 |
| Total | 90 | 100 |

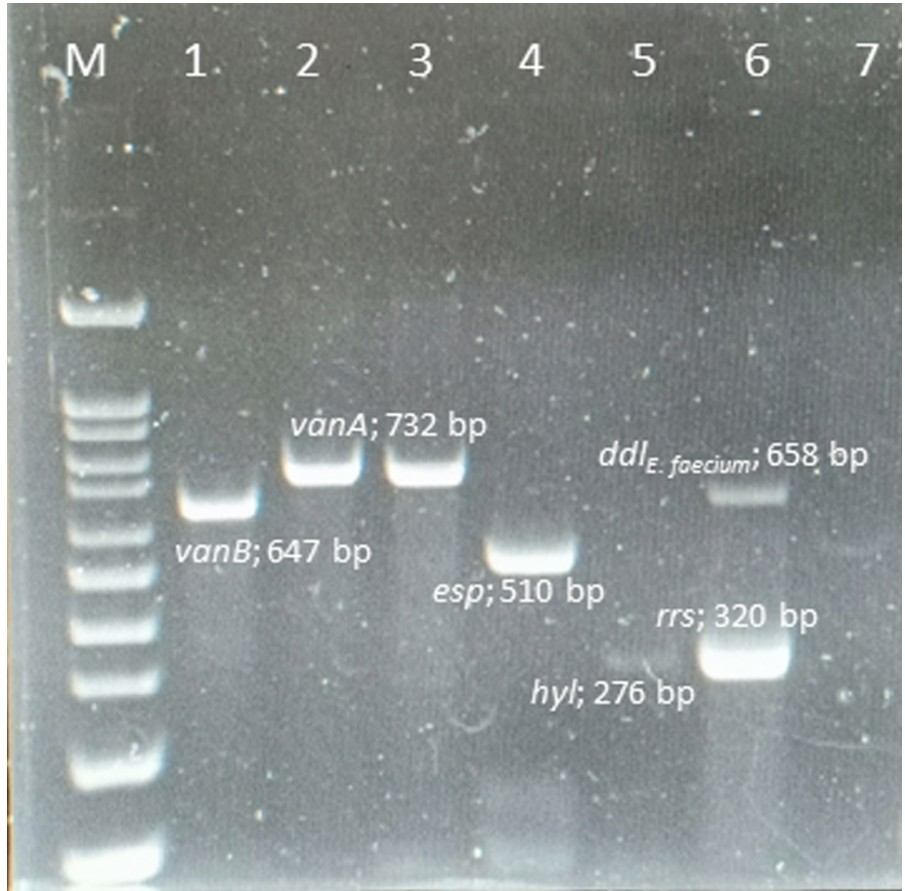

**Figure 1** **Multiplex PCR positive isolates.** Lane M = marker 100-1000 bp; lane 1 = *E. faecalis* ATCC 51299 *vanB* positive control; lane 2 = *E. faecium* ATCC 700221 *vanA* positive control; lane 3 = *vanA* positive from clinical strain *E. faecium* ; lane 4 = *esp* positive from clinical strain *E. faecium*; lane 5 = *hyl* positive from clinical strain *E. faecium*; lane 6 = *rrs* genes and *ddl*$_{E.faecium}$ from clinical strain; lane 7 = negative control .

isolates; and I–N genotypes, two isolates. Thirty-nine isolates were uniquely genotyped. MDR and XDR isolates were distributed among all scattered patterns, except for the three genotypes of the XDR pattern (J, L, and N) (Table 6). The heterogeneity among the isolates obtained from urinary tract infections was stronger than those from other infection sites.

## DISCUSSION

Enterococci significantly contribute to nosocomial infections. These infections most often occur in health care settings, particularly in intensive care units (ICU). They can spread by person-to-person contact or from contaminated medical devices (*Olawale, Fadiora & Taiwo, 2011*). Control measures to prevent the spread of enterococci infections are essential. To monitor the spread of enterococci infections at our hospital, samples collected from patients are routinely screened by conventional biochemical tests for nosocomial

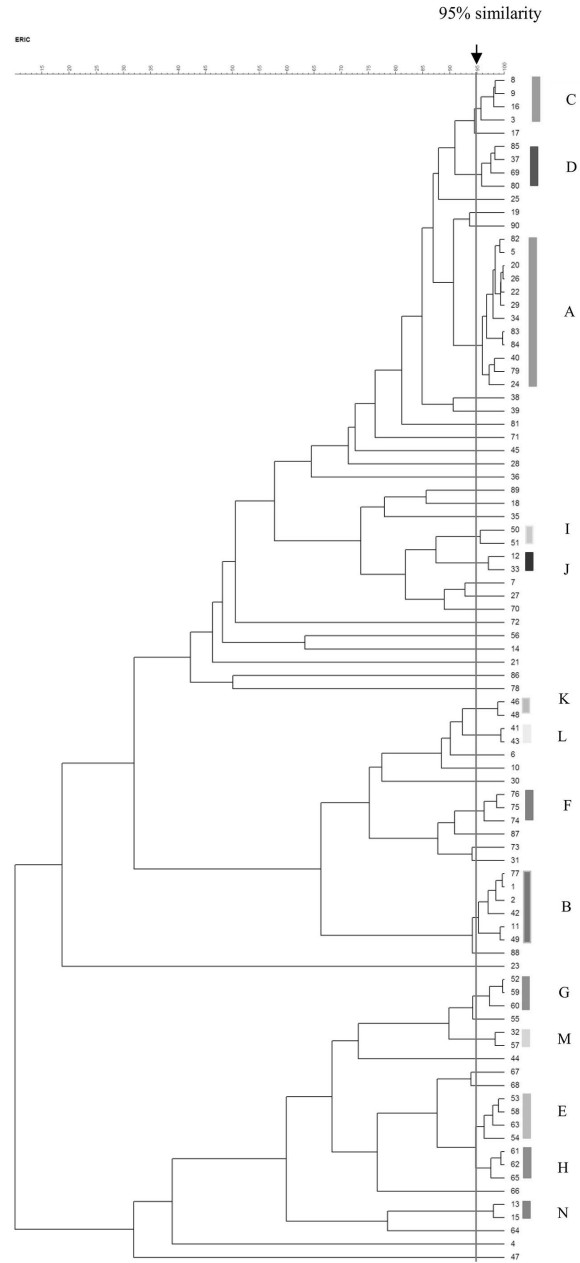

**Figure 2 Dendrogram of ERIC-PCR patterns among 90 VRE isolates.** Dendrogram of ERIC-PCR patterns showing the genetic relationship among 90 VRE isolates collected from clinical specimen in Songklanagarind Hospital, Thailand. Similarity > 95% were considered for clustering of isolates.

enterococci (resistant or sensitive to vancomycin). The prevalence of VRE among the nosocomial enterococci isolates in our study was 0.2%–4.3%, resulting in an average prevalence of 1.2%.

The prevalence of VRE isolates in our study was relatively low compared with that found in India (7.0%) (*Bhatt et al., 2015*) and in Egypt (6.3%) (*Said & Abdelmegeed, 2019*),

**Table 6   ERIC-PCR pattern, source, hospital collection location, antibiogram profile, and isolation date of multidrug-resistant (MDR) and extensively drug-resistant (XDR)-VRE strains from hospitalized patients at Songklanagarind Hospital, southern Thailand.**

| Code of isolates | ID | source | ward | Male (M) Female (F) | age | Collection date (day/month/year) | Resistance genes | Resistance patterns | ERIC types |
|---|---|---|---|---|---|---|---|---|---|
| 5 | 874-3-PSU | rectal | surgical ward | M | 70 | 05032014 | vanA, esp | MDR | A |
| 20 | 1949-PSU | rectal | medical ward | M | 60 | 28072011 | vanA, esp | MDR | A |
| 22 | 1114-PSU | blood | surgical ward | F | 64 | 07082014 | vanA, esp | MDR | A |
| 24 | 198-PSU | body fluid | surgical ward | M | 67 | 02082014 | vanA, esp | MDR | A |
| 26 | 1949-PSU | rectal | medical ward | M | 60 | 13082014 | vanA, esp | MDR | A |
| 29 | 1007-PSU | rectal | medical ward | M | 87 | 06082014 | vanA, esp | MDR | A |
| 34 | 4129-PSU | tissue | intensive care units | F | 53 | 28022012 | vanA, esp | XDR | A |
| 40 | 5056-PSU | urine | medical ward | F | 88 | 29032012 | vanA, esp | XDR | A |
| 79 | 5296-12br-PSU | body fluid | medical ward | M | 73 | 24122018 | vanA, esp | MDR | A |
| 82 | 2484-1u-PSU | urine | medical ward | F | 74 | 11012019 | vanA, esp | MDR | A |
| 83 | 7028-10u-PSU | urine | surgical ward | M | 70 | 30102018 | vanA, esp | MDR | A |
| 84 | 2079-4u-PSU | urine | medical ward | F | 44 | 11042018 | vanA, esp | MDR | A |
| 1 | 3044-PSU | rectal | medical ward | M | 73 | 17032014 | vanA, esp | MDR | B |
| 2 | 3576-PSU | rectal | surgical ward | M | 90 | 20032014 | vanA, esp | MDR | B |
| 11 | 3180-8-PSU | urine | medical ward | F | 38 | 19082015 | vanA, esp | MDR | B |
| 42 | 696ii-PSU | blood | medical ward | F | 63 | 04122013 | vanA, esp | MDR | B |
| 49 | 580-PSU | body fluid | intensive care units | F | 21 | 04022013 | vanA, hyl | XDR | B |
| 77 | 5296-PSU | body fluid | medical ward | M | 73 | 24122018 | vanA, esp | MDR | B |
| 3 | 3374II-PSU | blood | medical ward | F | 23 | 20022014 | vanA, esp | MDR | C |
| 8 | 3384-PSU | rectal | surgical ward | F | 71 | 19032014 | vanA, esp | MDR | C |
| 9 | 3576-PSU | rectal | surgical ward | F | 90 | 20032014 | vanA, esp | MDR | C |
| 16 | 2725-PSU | tissue | medical ward | M | 67 | 15032016 | vanA, esp | XDR | C |
| 37 | 1879-PSU | blood | medical ward | M | 86 | 10032011 | vanA, esp | XDR | D |
| 69 | 874-PSU | rectal | surgical ward | M | 70 | 05032014 | vanA, esp | MDR | D |
| 80 | 940-1u-PSU | urine | medical ward | F | 85 | 04012019 | vanA, esp | MDR | D |
| 85 | 5005-12u | urine | medical ward | F | 83 | 23122018 | vanA, esp | MDR | D |
| 53 | 762-PSU | body fluid | medical ward | M | 37 | 05022014 | vanA, esp | XDR | E |
| 54 | 176-PSU | blood | gynecology ward | F | 25 | 01112013 | vanA | MDR | E |
| 58 | 1243-PSU | MSU | gynecology ward | F | 70 | 08052013 | vanA, esp | XDR | E |
| 63 | 135-PSU | body fluid | operating room | F | 64 | 01052012 | vanA | XDR | E |
| 74 | 3784-PSU | rectal | medical ward | M | 46 | 21032014 | vanA, esp | MDR | F |
| 75 | 32-PSU | urine | medical ward | F | 79 | 01062012 | vanA, esp | XDR | F |
| 76 | 5552-1 I-PSU | blood | intensive care units | F | 0.25 | 24012019 | vanA, esp | MDR | F |
| 52 | 1134-PSU | rectal | gynecology ward | F | 52 | 07112013 | vanA, esp | MDR | G |
| 59 | 3236-PSU | MSU | medical ward | F | 50 | 18102016 | vanA, esp | XDR | G |
| 60 | 977-PSU | MSU | medical ward | F | 90 | 6102016 | vanA, hyl | XDR | G |
| 61 | 1630-PSU | body fluid | medical ward | F | 74 | 10092013 | vanA, esp | MDR | H |
| 62 | 4399-PSU | urine | surgical ward | F | 28 | 29042012 | vanA, esp | XDR | H |

**Table 6** (*continued*)

| Code of isolates | ID | source | ward | Male (M) Female (F) | age | Collection date (day/month/ year) | Resistance genes | Resistance patterns | ERIC types |
|---|---|---|---|---|---|---|---|---|---|
| 65 | 3093-PSU | rectal | gynecology ward | F | 52 | 18112013 | *vanA, esp* | MDR | H |
| 50 | 4215-PSU | urine | medical ward | F | 68 | 22042012 | *vanA, esp* | MDR | I |
| 51 | 5324-PSU | body fluid | surgical ward | M | 66 | 30082012 | *vanA, esp* | XDR | I |
| 12 | 3672-PSU | urine | surgical ward | F | 79 | 20022015 | *vanA, esp* | MDR | J |
| 33 | 1329-PSU | rectal | medical ward | M | 74 | 08082014 | *vanA, esp* | MDR | J |
| 46 | 4006-PSU | urine | medical ward | F | 73 | 24122013 | *vanA* | MDR | K |
| 48 | 3209-PSU | body fluid | medical ward | M | 66 | 19082012 | *vanA, esp* | XDR | K |
| 41 | 910-PSU | MSU | medical ward | F | 85 | 06082012 | *vanA, esp* | MDR | L |
| 43 | 2000-PSU | MSU | gynecology ward | F | 42 | 11112013 | *vanA* | MDR | L |
| 32 | 198-PSU | body fluid | surgical ward | M | 63 | 02082014 | *vanA, esp* | MDR | M |
| 57 | 5245-PSU | pelvic | operating room | F | 19 | 28092016 | *vanA, esp* | XDR | M |
| 13 | 3398-PSU | urine | surgical ward | M | 78 | 18092015 | *vanA, esp* | MDR | N |
| 15 | 4190-PSU | tissue | medical ward | F | 38 | 25042017 | *vanA, esp* | MDR | N |

**Notes.**
MSU, mid-stream urine.
BioSample accessions: SAMN18201951, SAMN18201952, SAMN18201953, SAMN18201954, SAMN18201955, SAMN18201956, SAMN18201957, SAMN18201958, SAMN18201959, SAMN18201960.

indicating that risks of VRE infections may be associated with geographic variability. *Blanco et al. (2017)* reported in 2017 that relative humidity and geographical location were factors affecting VRE colonization rates in adult patients admitted to ICUs in the United States. Increasing humidity resulted in a greater VRE colonization rate, and VRE colonization was lower in southern states. In developing countries, hospitals could inadvertently be increasing the risks of VRE colonization and infections because of the lack of oversight procedures for transferring patients from hospitals with VRE outbreaks (*Kang et al., 2014*; *Resende et al., 2014*).

Risk factors for severity of the patient's condition, hemodialysis, and central venous catheter usage and in univariate analysis include male gender, alcohol use, and antibiotics. Hospitalization in certain units of the hospital and lack of isolation measures are primary factors that affect VRE acquisition (*Fossi Djembi et al., 2017*). Our study demonstrated similar results after our 90 patients, whose test samples were positive for VRE, were paired with demographic and clinical data from the patient database. Nearly 50% of VRE carrier patients were ≥65 years old, with a median age of 64 years, and were admitted to the medical ward. In a 2019 study by Mathis et al., older patients had the highest infection during VRE outbreak in 2013–2014 at a teaching hospital in Lyon, France (*Mathis et al., 2019*).

In addition, positive VRE samples were detected in urine specimens (28.9%), followed by those obtained from body fluid (20.0%) and the rectal area (20.0%). These results were similar to a Brazilian study of VRE infection, where the highest VRE concentrations were isolated from patients with urinary tract infections (*Resende et al., 2014*). Vancomycin-resistant *E. faecalis* is a significant cause of nosocomial infections stemming from the

urinary tract, surgical wounds, and catheter use and affecting the bladder, prostate, and kidneys (*Zalipour, Esfahani & Havaei, 2019*).

Because VRE are multidrug-resistant organisms with limited therapeutic alternatives that result in increased morbidity and mortality in patients (*Buetti et al., 2019*; *Zalipour, Esfahani & Havaei, 2019*), we examined the 90 VRE isolates in our study for antimicrobial susceptibility. The findings showed resistance to $\beta$-lactam, glycopeptide, and fluoroquinolone antibiotics in most of the tested isolates, with the highest resistance rate to ampicillin (74.4%) and imipenem (62.2%).

Enterococci have intrinsic resistance to several classes of antibiotics that inhibit cell wall synthesis, such as $\beta$-lactams and glycopeptides, and they exhibit the ability to acquire new mechanisms of resistance, particularly to penicillin/ampicillin, aminoglycosides (high-level resistance), and glycopeptides (*Cetinkaya, Falk & Mayhall, 2000*; *Gagetti et al., 2019*; *Werner et al., 2008*). Results also showed susceptibility to meropenem in 10 of 15 tested isolates and 7 of the 21 tested isolates were susceptible to gentamicin. Meropenem and gentamicin are $\beta$-lactam and aminoglycoside antibiotics, respectively, and combinations of meropenem and aminoglycosides could function as alternative therapies for multi- and VRE infections. Similarly, there is evidence that $\beta$-lactams in combination with daptomycin reduced bacterial activity (*Smith et al., 2015*). Moreover, this combination of antibiotic therapy was successful in the treatment of endocarditis/severe MDR enterococcal infection (*Antony et al., 1997*; *El Rafei et al., 2018*).

*E. faecium* and *E. faecalis* are most frequently detected, and *E. durans*, *E. avium*, *E. casseliflavus*, *E. hirae*, *E. gallinarum*, *E. raffinosus*, and *E. muntdii* are less commonly detected (*Agudelo Higuita & Huycke, 2014*; *Zaheer et al., 2020*). In this study, all the detected VRE isolates were identified as *E. faecium*, a species rarely found in Songklanagarind Hospital. Other studies have found that vancomycin-resistant genetic elements can transfer between enterococcal species (*Murray, 1998*). Most VRE isolates were MDR, and only six VRE isolates were extensively drug-resistant (XDR). All VRE isolates in this study were *vanA E. faecium* strains, whereas *vanB*, *C*, *D*, and *E* were not detected, similar to the findings from a 2014 report from (*Resende et al., 2014*).

Enterococcal virulence that results in pathogenicity is considered a multifactorial process (*Comerlato et al., 2013*). Virulence factors promoting the emergence of hospital-acquired enterococcal infections have been reported in a number of studies. In our study, VRE strains originally isolated from urine specimen expressed a high frequency of the *esp* gene (91.1%), confirming the important role of *esp* in enterococcal colonization and biofilm formation in urinary tract infections (*Comerlato et al., 2013*). This finding was comparable to that of other investigators who reported that the incidence levels of *esp* in *E. faecium* isolates were 71.5% in northwest Iran (*Sharifi et al., 2012*) and 90% in China (*Yang et al., 2015*) but rather low in southwest Iran (57.5%) (*Arshadi et al., 2018*).

Other virulence factors, *hyl*, *asa*, *gel*, and *cyt* genes, were also analyzed. The *hyl* gene was detected in only five (5.6%) VRE isolates in our study, similar to the findings from Haghi et al. who found the *hyl* gene in 3 of 79 (3.8%) VRE isolates in northwest Iran (*Haghi, Lohrasbi & Zeighami, 2019*). We did not detect the *asa*, *gel*, and *cyt* genes, confirming that

*esp* and *hyl* are primary virulence factors among clinical strains of *vanA E. faecium* and play an important role in pathogenesis.

Molecular techniques for typing bacteria can vary when comparing standardization, cost, reproducibility, discriminatory power, and interpretation. We found that ERIC-PCR is a straightforward procedure for distinguishing organisms that are closely related. This method is less expensive and faster than many other methods, such as PFGE and amplified fragment length polymorphism typing (*Borgmann et al., 2007*).

The ERIC-PCR results identified 14 clusters among the 51 VRE isolates. The majority of the isolates were clustered in the A genotype, mostly from urine samples, and collected from the medical ward, and these were resistant to ampicillin, gentamicin, and carbapenem, similar to a report from Columbia (*Corredor et al., 2019*). We identified 39 unique traits of VRE isolates in our study, implying a circulation of these strains in the Songklanagarind Hospital for a period of time prior to the present study. These findings may suggest a hospital epidemiology stemming from the spread of MDR and vancomycin-resistant *E. faecium* with endemic *vanA* VRE. These bacteria may contribute to antibiotic resistance and virulence through horizontal gene transfer.

## CONCLUSIONS

This was the first study conducted in a tertiary care hospital in southern Thailand to show a high prevalence of MDR and XDR-VRE in patients, which could lead to pathogenic effects on public health. The results of this study suggest that overused antibiotics appear to be mediated mainly by the *vanA* gene, which carries one or two virulence genes that create a high resistance level to vancomycin. However, molecular techniques are highly effective in detecting antibiotic-resistant genes, which would allow monitoring to control VRE infection in hospitals.

### Funding

This work was supported by the government budget of Prince of Songkla University, with a grant number of MED601339S. The funders had no role in study design, data collection and analysis, decision to publish, or preparation of the manuscript.

### Grant Disclosures

The following grant information was disclosed by the authors:
Prince of Songkla University: MED601339S.

### Competing Interests

The authors declare there are no competing interests.

### Author Contributions

- Phanvasri Saengsuwan conceived and designed the experiments, performed the experiments, analyzed the data, prepared figures and/or tables, authored or reviewed drafts of the paper, and approved the final draft.

- Kamonnut Singkhamanan performed the experiments, authored or reviewed drafts of the paper, and approved the final draft.
- Siribhorn Madla analyzed the data, prepared figures and/or tables, authored or reviewed drafts of the paper, and approved the final draft.
- Natnicha Ingviya performed the experiments, authored or reviewed drafts of the paper, sample preparation, and approved the final draft.
- Chonticha Romyasamit performed the experiments, authored or reviewed drafts of the paper, did the quality appraisal, data collection, and approved the final draft.

## Ethics

The following information was supplied relating to ethical approvals (i.e., approving body and any reference numbers):

The study was approved by the ethical committee of the Faculty of Medicine, Prince of Songkhla University (REC-60-234-04-7).

## DNA Deposition

The following information was supplied regarding the deposition of DNA sequences:

The sequences are available at GenBank:

BioProject ID: PRJNA707344

BioProject ID: PRJNA707345

BioProject ID: PRJNA707346

BioSample accessions SAMN18201951–SAMN18202001

## Data Availability

Our SQL file is available in the Supplementary Files.

## Supplemental Information

Supplemental information for this article can be found online at http://dx.doi.org/10.7717/peerj.11478#supplemental-information.

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
