# Peer review of "Molecular epidemiology of vancomycin-resistant Enterococcus faecium clinical isolates in a tertiary care hospital in southern Thailand: a retrospective study"

_PeerJ, doi:10.7717/peerj.11478_

## Round 0.1 · original submission · Major Revisions

Your manuscript is evaluated by three reviewers. All reviewers find the work important but have raised some issues. Please make changes according to the reviewer's comments and or answer in the rebuttal.

Reviewer 1 ·

Basic reporting

The grammar needs to be corrected

Experimental design

The results are supported by the analysis

Validity of the findings

Conclusions and discussion should be expanded. More specifically, practical guidance (even though speculation) should be provided for monitoring VRE infection.

Additional comments

This is an interesting manuscript by Saengsuwan et al that studies the molecular epidemiology of vancomycin-resistant Enterococcus (VRE) faecium in southern Thailand hospitals. The authors show that molecular techniques can detect antibiotic resistant genes and inform monitoring techniques for VRE. This manuscript is a step forward in the framing of policies for VRE monitoring. It is written in a very simple and easy to follow manner. The conclusions are supported by the analysis. There are some minor concerns that need to be addressed before publication at PeerJ

1) Medical history of the patients- past drugs etc should be mentioned if any
2) The authors should aim to run the analysis on some other populations apart from Thailand. In its current form, there is a huge population bias and the results may not be generalizable. Further more, the results should be contrasted with (Bhatt et al. 2015b) and (Said & Abdelmegeed 2019)
3) Please expand this discussion more “Meropenem and gentamicin are β-lactam and
aminoglycoside antibiotics, respectively, and combinations of meropenem and aminoglycosides might be alternative therapies for multi- and VRE infections.”
4) Is there a correlation between age and VRE infection?
5) The figure legends should be expanded
6) Please check for grammar

Reviewer 2 ·

Basic reporting

- The authors have screened a decent number of nosocomial enterococci (~9000), of which 90 isolates were VRE and therefore selected for the study. However, these are important numbers that should be mentioned in the abstract and main text.

- Table 4 heading needs to be corrected. In addition to the List of antibiotics used, the table also contains other details that should be mentioned in the heading.

- All the tables are numbered as table 1 (Line 1 of pages 26,28,30,32,36,38)

-Figure 1: The figure legend mentions lane 6 = ddl genes and vanB positive from a clinical strain, but in the figure the bands are labeled as 16SrRNA and E.faceium. The authors need to correct this.

Experimental design

Method section: Detection of van genes and virulence genes is mentioned twice (Line 134-157)

Validity of the findings

The authors mentioned that year 2014 had the highest number of VRE isolates which is correct but the reason for its selection is wrong. The authors compared the number of VRE isolates in 2014 (28) to the total number of VRE isolates i.e. 90. The authors need to actually compare the VRE(%) in table 1. The fact that the year 2014 had the highest percentage (4.3%) of VRE positive strains makes this year with the highest prevalence of VRE.

The same thing also applies while comparing other characteristics. For eg., when comparing the gender ratio, the number of VRE isolates is higher in females than in males, and this may be because of more sampling from females compared to males.

Additional comments

The current study monitors the prevalence of drug-resistant Enterococcus faecium in a tertiary care hospital in Southern Thailand. This is an important study that is well performed and written and the results are discussed in detail.

Comments:
- Antimicrobial resistance profiles: The authors need to explain why all the 90 VRE isolates were not tested against different antibiotics
- As mentioned in the above section, the authors need to also consider the total number of isolates when comparing different characteristics such as year, gender, age, source, and hospital unit

·

Basic reporting

The paper is well written, straightforward, and easy to read.

Experimental design

'no comment'

Validity of the findings

Conclusions are consistent with the results presented.

Additional comments

The current manuscript reports the surveillance of vancomycin-resistant enterococci (E. faecium and E. faecalis) from a hospital in southern Thailand collected for eight-years. Authors find >50% VRE strains were isolated from >65 yr old. They found VRE samples were detected highest in urine specimens (28.9%) among all the specimens collected. Multiple-PCR and Genotyping study with ERIC-PCR finds that these strains contain a high frequency of esp gene, necessary for colonization and biofilm formation and vanA gene, reported in the MDR and vancomycin-resistant E. faecium.

The authors present the surveillance study of VRE in a hospital setting in Thailand, which offers essential health concerns for a developing country. The paper is well written, straightforward, and easy to read. Conclusions are consistent with the results presented. I will recommend the report to be accepted in its current form.
- Some comments:
- Authors can submit MDR strains reported in the manuscript to NCBI.
-Thank you for providing the raw data. However, your supplemental files need more descriptive metadata, and it will be useful for readers and your NCBI submission. For. e.g., Strain name, isolated from which specimen and genotyped, raw sequence.

---

## Round 0.2 · accepted · Accept

Your manuscript is reviewed by two reviewers. Based on your reviewer's comments, we accept your manuscript for publication in PeerJ.

Reviewer 1 ·

Basic reporting

Good

Experimental design

solid

Validity of the findings

Good

Additional comments

Congratulations on a good manuscript.

·

Basic reporting

The manuscript meets the criteria of reporting.

Experimental design

The manuscript meets the criteria of PeerJ.

Validity of the findings

Please see the General comments sections below.

Additional comments

This study by Phanvasri Saengsuwan et al investigates the prevalence of virulence and antibiotic resistance genes from a tertiary care teaching hospital in Thailand. The authors have utilized molecular biology techniques (including Multiplex PCR and ERIC-PCR) and Antimicrobial susceptibility assays to analyze the prevalence of resistance genes in patient samples. Overall, it is an interesting study and certainly useful to monitor VRE infection in hospitals. The manuscript can be accepted in the current format for publication in PeerJ. However, the authors should be more careful before drawing the comparison between male and female participants (line number 185), since these numbers are not normalized against the total number of male and female participants (out of 9273 participants mentioned in Table 1). It is also recommended to test all the VRE isolates against all different antibiotics, this would enable more meaningful conclusions to be drawn.